# Uses and Abuses of Religion in the Contemporary Legal Development of Montenegro: Undermining the Principle of Secularity

**Aleksandra Vukasinovic** [1,*] **and Biljana Damjanović** [2]

1    Constititional Court of Montenegro, 81000 Podgorica, Montenegro
2    Law Faculty, Mediterranean University, 81000 Podgorica, Montenegro
*    Correspondence: alleksandrav@icloud.com

**Abstract:** In the context of contemporary legal and political development, this work aims to analyze, through the prism of the constitutional principles of secularism and the civic state, the growing influence of religion on politics in Montenegro which has indirectly caused tectonic changes in the current legal relationship between the state and religious communities, at least temporarily questioning these everlasting values of constitutional democracy. Our basic hypothesis is that, in this correlation between the state and the Orthodox Church, religion is being used not only as a belief system which primarily belongs to the spiritual sphere of individuals, but also as a tool in redefining national identity and achieving politically desirable results at the public level. In the interpretation of positive legal regulations, this paper predominantly uses teleological and normative methodology together with sociological and axiological methods, necessary for understanding the broader context of the widened scope of bonding between religion and politics. This paper is also accompanied by relevant literature, which is promising in terms of solving the very interesting issues which once belonged to the "spirits of the past" and yet in the twenty-first century have been modernized at the place where the internal legal order of Montenegro and the Orthodox Church meet.

**Keywords:** national identity; constitutional identity; state-church relationship; intertwining religion and politics; secularism





## 1. Introduction

Every modern legal community in Europe accepts a certain system of common constitutional principles (democracy, freedom, rule of law, secularity). From the point of view of this system of values, the legal relationship between the state and religious communities can be assessed as positive or negative, desirable or undesirable, prone to use or abuse of religion for political purposes. The position of political leaders on various religious issues during the electoral process has become an information framework for assessing their reliability and, ultimately, for electing them (Service 2020, p. 2). The intertwining of religion and politics, the legal order of the state and the church, and their coexistence and emphasized interdependence are especially evident in Montenegro from 2020 to the present day. Montenegro, *de facto* and *de jure*, is becoming a unique example in postmodern Europe in which the Orthodox Church has directly influenced the outcomes of parliamentary elections that have resulted in the institutionalized ending of the 30-year regime of one-party dominance. Indirectly, this has caused tectonic changes in the legal relationship between the state and religious communities, at least temporarily questioning the civic and secular constitutional character of the state.

The question of setting boundaries between the state and religious communities is not at all easy, especially in societies deeply burdened with identity issues, in which there is a fine line between the enjoyment of a right and its abuse. The significance of religion is most visible in post-socialist regimes, where religion, along with language and state symbols,

becomes one of the pillars of the new identities of states created by the disintegration of communist systems. This attitude is often combined with "accusations" against Orthodox churches that they are "nationalist institutions," in which "the ideology of the nation is reinforced by the theology of the nation." This thesis was also synthesized by Vjekoslav Perica, who defines Orthodoxy as a religion in which "the church, ethnic community and state grow together" (Perica 2002, p. 7). The interdependence of religion and politics, although current, is immanent not only in transitional societies but also in societies and states with a high degree of democracy. For example, a poll was conducted in the United States on the influence of religion and race on political orientation towards Republicans and Democrats, where the survey showed a 12% shift in favor of the Democrats in the population of white evangelicals who had previously voted Republican (Jones 2019, p. 50).

We must be aware of the fact that we cannot predominantly limit the issue of religion to the question of understanding other or different religions, but it is necessary to pay special attention to how to experience one's own religion and develop self-criticism in light of all modern social challenges (Aslan 2019, p. 437). This is precisely the aim of this paper, in the context of contemporary constitutional legal developments in Montenegro, to shed light on the relationship between the state of Montenegro and the Orthodox Church through the prism of the constitutional principle of secularity and the common European values of multiculturalism and tolerance. Our basic hypothesis is that the civic and secular constitutional identity in Montenegro is being shaken by different concepts and views of *demos*, the past and future of the community, and increasing polarization within the dominant Orthodox population, the articulation of which was most apparent in the pre-constitutional period, yet which is becoming modernized today at the spot where the internal legal order of Montenegro and the Orthodox Church meet. Religion in this relationship is not only being used as a belief system, which is its main purpose, but we will also talk about its abuse in creating politically desirable results.

## 2. The Importance of Religion in the Construction and Reconstruction of Montenegrin National Identity

The concept of identity is very complex, often disputed, and ambiguous (Mihovilovic and Boulton 2020, p. 9), consisting simultaneously of social and individual, equality and diversity, theory and practice, permanence and progress, staticity and dynamism (Clarke 2009, p. 189). The construction of national identity, in the broadest sense, implies the process of shaping individual cognition as belonging to a particular collectivity based on the common values of "collective enthusiasm," such as language, culture, and religion (Seton-Watson 1918). Understanding the impact of these values on the creation of groups (which are by no means immutable) reveals the processes, frameworks, and categories through which ethnicity and nation are constantly re-shaped from within (Džankić 2014). Brubaker's thesis that national identity represents only a temporary representation of collectivity, the framework through which the united ascribe themselves to groups (Brubaker 2004), is particularly evident in the modern development of the Montenegrin national being, characterized by a significant change in the perception of the nation and the reconstruction of what it means to be a Montenegrin.

There is no doubt that one of the main characteristics of the use of religion for political purposes in Montenegro is national identity. However, it does not follow from this that this relationship has a religious origin, but that religion in Orthodoxy is politically used as a means of homogenization of a particular ethnic group. History proves that, if we use religion as a means of political division, the result can very easily, according to Hannah Arendt's famous theory, be the transformation and perversion of religion into an ideology and the corruption of our struggle against totalitarianism by fanaticism that is completely alien to the very essence of freedom (Arendt 1953).

Interpretation of the contemporary coexistence of religion and politics in Montenegro is possible only by understanding the origin of this relationship, which is deeply rooted in the history and tradition in which the Orthodox Church has had primary importance

in the constitution, survival, and organization of the Montenegrin state and by analyzing its place in the modern order of the state shaped by the normative-legal framework of the community, which should be followed in order to resolve any disputes arising from this very sensitive relationship in a manner appropriate to a politically mature and civilized society.

Until the middle of the nineteenth century, there was a unity of ecclesiastical and secular government in Montenegro, in the form of a prince-bishop. All important legal documents from the earlier phase of Montenegrin statehood indicate that the legal consciousness of Montenegrins was largely based on a religious code, and that the Orthodox Church created the state, and not vice versa (Bogićević et al. 2002). Disputes arise as to whether this Orthodox Church was an autocephalous Montenegrin one, or whether it belonged to the jurisdiction of the Serbian Orthodox Church. There are historical sources that support the arguments on both sides, but in this paper we will not deal deeper with this issue. It is necessary to emphasize that the loss of Montenegrin independence in 1918, followed by the political unification of the southern Slavs into the Kingdom of Serbs, Croats, and Slovenes, was accompanied by the religious unification of the existing autocephalous and other Orthodox churches, including the Montenegrin one, into a unified Serbian Orthodox Church.

After the disintegration of Yugoslavia, the transformation of national identities on this soil began. Consequently, Montenegro has built the genealogy of its modern national identity in conditions of frequent social changes and in various socio-political systems, which have caused, as Đuro Šušnjić rightly notes, an identity crisis "with the awareness of individuals that there are several elements of personality in each individual who are struggling for supremacy—and they do not agree with each other" (Šušnjić 2009, pp. 405–21). Throughout history, the national identity in Montenegro has been dualistic—Montenegrin and Serbian—they were not mutually exclusive categories, a situation that is interestingly described as the Montenegrin *homo duplex* (Darmanović 1992, p. 28). This is evidenced by the empirical data of national censuses in Montenegro over the past half century, which confirm that the division between Montenegrins and Serbs is a matter of the individual changing their individual feeling and political commitment, and not of real identity differences, if we assess identity with the common values of culture, religion, and language.

Two nations with the same origin, the same language, and the same religion begin to share exactly what they have in common, so religion thus becomes a means of "polarization" and "political manipulation" within the national being of Montenegro. A phenomenon that can be described as a national fission, i.e., political splitting of the same identity core into two polarities, two nations.

Jenne and Bieber claim that Montenegro, not only at this moment but also over the past six decades, has been practicing so-called situational nationalism in which their national identities are changing in response to geopolitical battles that call for individual loyalty at the local, national, and international levels (Jenne and Bieber 2014). In times of political upheaval and elections, competition between these identities becomes more intense, leading to significant identity changes and sometimes political splits where there were none before.

Along with the development of the idea of Montenegrin independence, as well as the realization of this in the 2006 independence referendum, the need of the newly formed state to create a new identity, which would be in clear discontinuity with the previous orders in which Montenegro did not act in the full capacity of sovereignty, was born. The identity transformation was mostly related to the desire of the majority of, but not all, citizens to create a clear differentiation between the Montenegrin and Serbian national schemes, and its normative completion through the adoption of a new Constitution, in which the most representative reflection of these differences are the provisions on the Montenegrin language as the official and state symbols in Montenegro. The secession of Montenegro from Serbia and reconstruction of its own identity through normative constitutional regulation of provisions on the new flag, language, and new alphabet did not include the normative completion of the reconstruction of the position of religious

communities in the state, but in accordance with the principle of secularism established the principle of "render to the church what is the church's and to the state what is the state's".

In 2006, the secular authorities were separated into Serbia and Montenegro, but spiritual authority in Orthodoxy still belongs to the unified Serbian Orthodox Church, so it becomes the subject of political dispute that the church is in the service of spreading the Serbian identity in Montenegro, that it is anti-systemic because it is in the service of the political interests of another state, and that this prevents the complete rounding-off of the Montenegrin national being, denying the Montenegrin Orthodox population the right to restore the autocephaly of their religious community, which recognizes their ethno-national identity. This is all the more so since churches in Orthodoxy carry a national signifier in their name and have been criticized as nationalist institutions.

Daniela Kalkandjieva, in her study "*Comparative Analysis on Church-State Relations in Eastern Orthodoxy*," claims that the arguments for this thesis are limited in many respects. They mainly concern Eastern European countries in the most recent two centuries of their development. It is impossible to turn a blind eye to the temporal asymmetry between these two phenomena: nationalism is modern, while Orthodoxy has a much longer history. In a theological sense, Orthodox Christianity is open to all human beings regardless of their ethnicity, citizenship, social status, or gender. From the ecclesiastical point of view, Orthodox canonical law does not associate the establishment of an Orthodox church with a specific ethnic group or nation, but with a certain patriarchate and the limits of its territorial jurisdiction (Kalkandjieva 2011, p. 595). However, it often happens that the secular authorities do not always or completely respect the ecumenical principles on which the religious order is based, so in the provisions of Article 11 of the Law on Churches and Religious Communities of the Republic of Serbia it is stated that the Serbian Orthodox Church has an exceptional historical, state-building, and civilizational role in shaping, preserving, and developing the identity of the Serbian people.

Due to different national identity settings in Montenegro, religion in Montenegro is beginning to appear as a by-product of politics, and not just as a religious belief in itself. This is evidenced by the attempt to politically form the canonically unrecognized Montenegrin Orthodox Church in 1993, described as an "epiphenomenon of Montenegrin politics" (Ramet and Pavleković 2005), which has been functioning for years with very little support from the public. The citizens of Montenegro, years ago, showed much greater trust in religious communities than in state institutions and political parties, which implies that religion nevertheless transcends secular ideologies. According to independent surveys of public opinion, the Serbian Orthodox Church is the institution in which Montenegrins have the most confidence, and this is where the power of religion in creating politically relevant results comes from.

## 3. Undermining the Principle of Secularity

With regard to the constitutional relationship between the state and religious communities, the current Constitution of Montenegro prescribes a clear separation of religious communities from the state and their equality and freedom in the performance of religious ceremonies and religious affairs.

Provisions from Article 14 of the Constitution of Montenegro 2007[1] prescribe a clear separation of religious communities from the state: Religious communities shall be separated from the state. Religious communities shall be equal and free in the exercise of religious rites and religious affairs.

The creators of the constitution emphasized the normative importance of the principle of secularity for the functioning of the entire constitutional system by positioning it in the First Part of the Constitution named Basic provisions, but its real significance and scope will mostly depend on the constitutional interpretation of this constitutional provision in practice.

The Constitutional Court of Montenegro, as the only State organ responsible for the protection of constitutionality and legality, so far has not had a rich practice when it comes

to the interpretation of this constitutional principle. The only case in which it dealt with this principle, indirectly, is related to taking a position on the secularity of teaching in public schools. The position taken in the Decision of the Constitutional Court of Montenegro on the assessment of the constitutionality and legality of the Agreement on the regulation of relations of mutual interest between the Government of Montenegro and the Islamic Community in Montenegro, U-II no. 56/14 of 24 February 2017 stated that secularity of education and upbringing and the prohibition of religious activity refer to the content of the educational or publicly valid educational program which is implemented in public institutions that are not licensed as secondary religious schools, but does not refer to the expression of religious feelings of students who are receiving education based on those programs.

With the change in the political scene in the last two and a half years, the number of cases before the Constitutional Court in which a decision will have to be made on a possible violation of the principle of secularity has increased significantly. The strong participation of the Serbian Orthodox Church in contemporary politics of Montenegro began in the middle of 2019, when the Proposal of the new Law on Freedom of Religion was announced, which would require religious communities to prove ownership of buildings built before 1918. The Serbian Orthodox Church experienced this law as discriminatory because it potentially threatened its right to property.

Political tensions grew even more when, at the party congress, the President of the State of Montenegro and the Democratic Party of Socialists, Mr. Milo Djukanovic, which had been in power for 30 years at the time, announced that he would restore the autocephalous Montenegrin Orthodox Church. Everything culminated in the significant participation of the Serbian Orthodox Church in the 2020 election campaign, which resulted in the first change of government in an election after three decades.

These political disputes, which are rooted in national identity origins, inevitably influenced and still are influencing the legal framework of the relationship between the state and religious communities in Montenegro. This thesis is also confirmed by numerous attempts to resolve these political disputes through a constitutional dispute, by initiating a series of proceedings before the Constitutional Court, which, among other things, emphasize the potential violation of the principle of secularity.

In the past two and a half years, numerous of cases were launched before the Constitutional Court of Montenegro in this regard: an Initiative for the assessment of the constitutionality and legality of the Law on Freedom of Religion from 2019; Initiative and proposal for the assessment of the constitutionality and legality of the amendment to the Law on Freedom of Religion from 2021; The procedure of whether the President of the State violated the constitutional principle of secularity with his public statement about request of moving the place of enthronement of the Metropolitan of the Serbian Orthodox Church in 2022; Three initiatives for the evaluation of the constitutionality and legality of the Fundamental Agreement between Montenegro and the Serbian Orthodox Church in 2022/23. The Constitutional Court of Montenegro, due to a lack of quorum, has not yet decided on these cases.

Whether this type of ever-growing interweaving of politics and religion would constitute a violation of the principle of secularity is a matter of constitutional interpretation. For this purpose, it must be taken into account that the Constitution of Montenegro from 2007 expressly guarantees the commitment to a secular state. This commitment is legally more visible when compared with the solution in the previous Constitution of Montenegro from 1992 which stipulated the following in Article 11 named Religion: "The Orthodox Church, Islamic Religious Community, the Roman Catholic Church and the other faiths shell be separated from the state. All the faiths shell be deemed to be equal and free in the performance of their religious rites and affairs. All the religious denominations will independently arrange their interior organization and religious affairs within the legal set up. The State shell offer material assistance to religious denominations."

Unlike the 1992 Constitution of the Republic of Montenegro, which had recognized different religious communities, the Constitution of Montenegro 2007, in accordance with the civic character and the principle of secularity of the state, took a more neutral stance in relation to religious communities. Thus, the provision of Article 11 of the 1992 Constitution of the Republic of Montenegro, which allows the state to provide material assistance to religious communities, was intentionally omitted in the current Constitution.

Although the fields in which the legal order of the state and religious communities meet are numerous, setting the boundaries of this relationship will mostly depend on the interpretation of whether the principle of secularity in the Montenegrin constitutional system rests on strict separation or cooperative separation of the state and religious communities.

The system of strict separation implies the absence of any organic or functional links between religion and the state: state authorities do not support the work of religious communities (in financial or other terms), the legal order does not suffer the influence of any religious system, and the private and public spheres are strictly separate, whereby the expression of religious commitment is limited to private life. (Marinković 2011, p. 377). On the other side, the system of cooperative separation of the state and religious communities is also based on the principle of secularity, but it also implies the recognition of many tasks on which church and state cooperate. This cooperation implies, for example, that the state materially helps religious communities or that religion is studied in public schools.

When it comes to relations between religious communities and state jurisprudence, the European Court of Human Rights establishes a wide margin of appreciation for the state. (Case of Eveida v. Uk, number 48420/10, 59842/10, 51671/10 i 36516/10, from 27 of May 2013, paragraph 84). There is a freedom of each contracting state according to the European Convention to develop the most diverse relations between religious communities and the state, including the free choice of the mentioned systems relations, as long as it does not call into question the democratic character of a society based on pluralism, tolerance, and freedom of thought.

The jurisprudence of the Constitutional Courts of the Republic of Croatia and the Republic of Serbia, which share similar constitutional provisions on the principle of secularity as well as a similar legal heritage as Montenegro, is somewhat richer. The Constitutional Court of Republic of Croatia in its case U-II/2050/2011 interpreted the principle of secularity as a two-way barrier, as a principle that protects the autonomy of the religious community from state encroachment, but which at the same time has the task of preventing the interference of religious organizations in state affairs.

The Constitutional Court of Republic of Serbia in a very extensive case assessing the constitutionality and legality of the Law on Churches and Religious Communities Uz-455/2011, held the position that their Constitution of 2006 did not opt for a system of strict separation of church and state.

Namely, the Constitutional Court, proceeding from the provision of Article 11 and Article 44 para. 1 and 2 of the Constitution, which established that the Republic of Serbia is a secular state, and that churches and religious communities are equal and separate from the state and that no religion can be established as a state religion, stated that the aforementioned constitutional provisions, in themselves, do not mean a system of complete separation of church and state, but that there is no state church and that there is no identification of the state with a particular religion or religion in general, and that churches and religious communities are free to independently determine their internal organization and religious affairs and that the state must not hinder the adoption and implementation of autonomous regulations and decisions.

Therefore, the Constitutional Court of Serbia have established that in Republic of Serbia there is no system of strict separation of church and state, i.e., that one cannot speak of absolute separation of church and state, but of a system of cooperative separation.

Although, as we have seen, the Constitution of Montenegro from 2007 implies a more neutral relationship between the state and religious communities than the Constitution from 1992. This may indicate the intention of the constitution maker to prioritize a principle

of their strict separation, especially bearing in mind the civic character of the state. Yet on the other side, the objective fact is that the Constitution nowhere explicitly prohibits the establishment of relations between the state and religious communities on the principle of cooperation.

Reality shows that the interpretation of the principle of secularity is not only a product of the text of the Constitution itself, but also of social and political circumstances. This is not surprising, because the interpretation of the Constitution is also a social and cultural process (Haberle 2002, p. 34).

The experience of the Republic of Bulgaria also testifies to that fact that Montenegro is not an isolated case in the growing interference between the Church and State. According to Ina Merdjanova, even though the Constitution of Bulgaria recognizes the principle of secularity, the State of Bulgaria has continued to interfere in church affairs in contradictory ways; for example, in the 1990s it registered a second Synod and in 2004 it outlawed this Synod (Merdjanova 2022, p. 3).

Nevertheless, we conclude that regardless of whether the Constitutional Court of Montenegro decides to interpret the principle of secularism as a strict or cooperative separation of the state and religious communities, it is of essential importance to set clear boundaries in a satisfactory manner in order to preserve the legacy of the state's secularity.

The constitutional character of the state of Montenegro is based on the principle of secularity, that is, the principle of the separation of religious communities from the state, and this is the prevailing constitutional model of relations between the state and religious communities in Europe. The separation of state and church represents one of the most important advances in human history and it is also the most suitable institutional arrangement that enables tolerance, coexistence, and democracy within the framework of the state (Simović and Simeunović-Patić 2017, p. 114). In principle, this model implies the "secularity of the state," that is, the absence of a constitutionally guaranteed state church or state religion. However, the principle of secularity (laity) does not mean the absolute separation of religious communities and the state, i.e., relations in which there are no points of contact between them, but rather a principled division in which religious communities are free to perform religious affairs and religious ceremonies, and the state is sovereign in performing its powers that refer to the secular level (Vukčević 2021, pp. 82–84).

The principle of secularity is adapted to the specific social needs and demands of the state and society, and therefore on European soil we find several sub-models that are used in the normative regulation of the relationship between the state and religious communities: strict separation of the state and religious communities, as is the case in the Czech Republic, the Netherlands, Slovenia, Latvia, Estonia, and Portugal; deep mutual penetration of religious concepts and the organization of the constitutional system, as is the case in Cyprus; and a cooperative system (Decision of the Constitutional Court of Serbia, Uz-III No. 455/2011, dated 16 January 2013) between two forms of social authority, as is the case in Bulgaria, Belgium, Serbia, and Spain (Petrov and Mikić 2015, pp. 625–48).

Regardless of the undoubted importance attached to the principle of secularity of the state, it can be noted that the contemporary European public space is marked by a discrepancy between the normative and the real (Simović 2017, p. 10) in the sense of a strict separation of the state and religious communities. Many constitutions, while standardizing the principle of secularity, do not set obstacles to establishing relations between the state and religious communities on the principle of cooperation. It is not unknown in practice for states to provide financial assistance to religious communities in various forms, to conduct dialogue with religious communities, to help in the construction of religious buildings, and to allow religious education in public schools, so in practice dilemmas often arise as to in which situations this separation must be strict and when there are opportunities for cooperation. States are left with the sovereign right to define this exclusively internal principle, and constitutional courts to give meaning and purpose to this principle in the constitutional order, respecting their own constitutional identity and the minimum standards set by the European Court of Human Rights.

The contrast between the clear secular constitutional provisions in Montenegro, on the one hand, and the different forms of state cooperation with religious communities and their political relevance in practice, on the other hand, reminds us of how difficult it is to assess the real nature of the relationship between religious communities and the state only on the basis of constitutional provisions. Adoption of the Law on Freedom of Religion or Belief and the Legal Status of Religious Communities (hereinafter: the Law on Freedom of Religion) opened the door for a new legal and political landscape of relations between the state and religious communities in Montenegro.

The first version of the Law on Freedom of Religion, adopted on 29 December 2019, caused heated disputes, mostly because of the provisions that affected the Serbian Orthodox Church and that related to church property and the registration of religious communities. The disputed provisions, among other things, stipulated that all religious buildings that had been the property of the state of Montenegro before the loss of its independence and annexation to the Kingdom of Serbs, Croats, and Slovenes, and which had not been legally transferred to the property of a religious community, would be recognized as state property (Article 62), as well as that unregistered religious communities acquire the status of a legal entity from the date of registration in the Register of Religious Communities kept by the competent ministry (Article 24 paragraph 1, Article 25 paragraph 3, and Article 28 paragraph 2):

> *"Religious communities that are reported and registered with the competent administrative body in Montenegro in accordance with the Law on the Legal Status of Religious Communities (Official Gazette of the Federal Republic of Montenegro 9/77) and operate in Montenegro on the day this Law enters into force are registered in the records of existing religious communities kept by the Ministry, by submitting an application for registration by a person authorized for representation."*

> *"A part of a religious community whose religious center is abroad and which operates in Montenegro can acquire the status of a legal entity in Montenegro by entering it into the Register or Records."*

> *"Unregistered and unrecorded religious communities do not have the legal status of religious communities that are registered or recorded in accordance with this Law and cannot acquire and exercise rights that, in accordance with the legal order of Montenegro, exclusively belong to registered or recorded religious communities, as legal entities."*

By prescribing (Article 24, paragraph 1 of the Law on Freedom of Religion) the possibility of entering into the records of existing religious communities only those religious communities that were registered and recorded according to the 1977 law, the Law on Freedom of Religion potentially discriminated against those religious communities that had existed and operated before the entry into force of the 1977 law, i.e., it discriminated against the Serbian Orthodox Church. Communities that had the status of a legal entity according to the 1953 law or the 1977 law, according to this law lose that right and they only have the option to register, as they actually become newly established religious communities. A question arose concerning what would happen to the property of a religious community which does not meet the legal requirement for registration and which operated on the territory of Montenegro before the entry into force of the Law on Freedom of Religion, decided to register and, according to the provisions of Article 18 paragraph 1, acquired the status of a legal entity.

Registration according to this legal solution had a constitutive character since a religious community is thus created and acquires the status of a legal entity by being entered into the Register of Religious Communities. Prescribing restrictive conditions for registration in the records of religious communities, i.e., prescribing the right to register without guarantees of legal and religious continuity, could have resulted in the relevant religious community losing property in that way (since the right to property is reserved only for registered and recorded religious communities) and in its being prevented from defending its interests before the court and the competent authorities or, if it were registered, it would

lose its legal and religious continuity, i.e., subjectivity, thereby also losing its property (because formally and legally it does not represent a legal successor of a religious community that had operated under the previous laws). These legal solutions have opened up a number of issues and led legislators into a zone where they are exceeding their powers and violating the principle of separation of the religious community from the state, as well as the principle according to which religious communities are equal and free in the carrying out of religious ceremonies and religious affairs, proclaimed by Article 14 of the Constitution of Montenegro, as well as the principle of freedom of thought, conscience, and religion proclaimed by the provision of Article 46 of the Constitution of Montenegro and the principles of Article 9 viewed through the prism of the rights guaranteed by the European Convention on Human Rights, contained in the provisions of Articles 11 and 14 of the Convention.

These controversial issues were resolved non-institutionally. They started a series of civic protests, better known as litanies, led by the Serbian Orthodox Church, and culminating in the parliamentary elections in August 2020 in the institutionalized ending of the 30-year period of dominance of the Democratic Party of Socialists in Montenegro because the Church consequently also stepped out of the zone of secularity and called on citizens to vote against the government and the parliamentary majority that had passed a law directly affecting their right to property, freedom of religion, etc.

One of the first moves of the newly formed government and the parliamentary majority was to change, i.e., delete, the disputed provisions of the law by adopting amendments in January 2021. In this way, the Serbian Orthodox Church, knowingly or unknowingly, has entered the political scene by the front door, influencing certain political and legal solutions, beyond the reach of the mere rights of religious communities. Conversely, public officials have increasingly begun to use religion in public discourse and support of the Church for their own political progress. In both cases, a shift away from the civic and a move towards the national was carried out, in which "the ideology of the nation is reinforced by the theology of the nation." Today, two and a half years later, there are numerous discussions on the topic of the secularity of Montenegro; some claim that the state is sliding towards theocracy, others indicate that religious freedom was being interpreted from a repressive point of view and that nothing unusual is happening. The fact is that issues related to religion are so present in the public sphere that they have directly or indirectly led to the overthrow of not one, but three governments since August 2020.

First, the Serbian Orthodox Church had a decisive influence in the fall of the 30-year regime of the Democratic Party of Socialists in the Parliamentary elections of August 2020. In this process, the Church proved to be the unifying factor of the people. It was precisely this unifying factor that was missing from the political parties that fought for the change of the regime. The strength of the Serbian Orthodox Church lies in the great trust it enjoys in public opinion, and hence its relevance at the political level. Second, after the fall of the 30-year regime, the negotiations for the formation of the new 42 government were conducted in Monastery Ostrog, with the presence of the Metropolitan of the Serbian Orthodox Church who tried to unite the winners of the elections. This government lasted only one year and fell after the cracks created by the events surrounding the enthronement of the new metropolitan of the Serbian Orthodox Church in Montenegro. Third, the formal reason for the vote of no confidence of the next 43 government, also after only one year, was the signing of the Fundamental Agreement between the State and the Serbian Orthodox Church. For the majority who voted no confidence in the 43 government, this Agreement is a series of provisions that violates the principle of secularity by giving the Serbian Orthodox Church the right to public powers, exemptions from coercive measures by state authorities, and the right to religious education, etc.

Despite the constitutional nonexistence of a dominant religion in Montenegro, practice shows that a religion can be perceived as dominant when it reflects the majority of believers in a certain country. Cesari makes the difference between a dominant, an established, and a hegemonic religion. (Cesari 2015, p. 1337) The manner in which open legal issues

between the state and the church will be resolved will also determine the understanding of the position and influence of the Serbian Orthodox Church on social, political, and legal processes in Montenegro. We should be aware of the fact that the dominant religion is slipping into a hegemonic one when the state grants certain religious group exclusive legal, political, or economic rights denied to other religions (Cesari 2015, p. 1337).

This powerful chain of political events that can lead to clericalization of society can be stopped only by returning the holders of political power and church dignitaries to the constitutional framework that is clearly based on principles of secularity and the civic state. Public space has become the scene of conflict between one political religion and another. Religious communities have their party preferences and openly play the political game. On the eve of the last presidential elections in Montenegro, held on 2 April 2023, the Serbian Orthodox Church in Montenegro called on the faithful to participate in the presidential elections.

Ethnicity, which usually but not necessarily in Eastern Orthodox countries comes with religious affiliation, is deeply rooted in the consciousness of our society. The presence of religious phrases in public discourse clearly shows the degree of undermining of the concept of secularity. All in all, it is clear more than ever that the civic concept of the state is defended, built, and preserved in such a way that space for the cooperation of all actors in the public sphere should be respected, but within the framework of the Constitution and the laws in the separate fields of activity of the state and church authorities.

The question of the relationship between the state and the church is not a question of the modern age, but a question that follows the development of humanity itself. From the Old and New Testaments, and the works of St. Augustine, through philosophers and theorists of the new age, to the present day, the basis of the distinction between the state and the church lies in the question of the moral and religious functions they accomplish (David 2023, p. 3). Thus, this relationship could be characterized as a relationship of peaceful interdependence (David 2023, p. 3), regardless of the tensions that often occur between the state and the church, which are an integral part of the enjoyment of their freedoms and rights. Maintaining a state of balance in the relationship between the state and the church is a prerequisite for the full affirmation of freedoms.

## 4. Distortion of the Concept of a Civic State

This newly emerging relationship, the increasingly close cooperation between the Serbian Orthodox Church and the Montenegrin state, calls into question both the civic and multicultural constitutional identity in Montenegro. According to the 2007 Constitution, the state was constituted, among other things, as a civic state, and a citizen who holds Montenegrin citizenship was designated as the holder of sovereignty. In a multinational community such as Montenegro, this concept has the potential strength to be an amalgam of civil democracy; however, the problem is that the attribution of power to the citizen, as the holder of sovereignty in post-authoritarian socialist regimes, represents a rewritten constitutional form, which in reality contradicts the real locus of power, which is actually carried by the dominant community (Zenović 2017, p. 16) or, in the case of Montenegro, the dominant communities.

On the other hand, there is no doubt that, as opposed to the civic one, the ethnocentric model of constitutional identity is the most prevalent one in all former Yugoslav republics. The current Constitution of the Republic of Croatia establishes Croatia "as the nation state of the Croatian nation and the state of the members of its national minorities." A similar provision exists in the Constitution of the Republic of Serbia: "The Republic of Serbia is the state of Serbian people and all the citizens who live in it." The Constitution of the Republic of North Macedonia in the Preamble states that Macedonia is constituted as a nation-state of the Macedonian people and the other nationalities living in it. The Constitution of Bosnia and Herzegovina, on the other hand, establishes a union with three other constituent peoples: Bosniaks, Croats, and Serbs, which, at least in principle, excludes the possibility of an individual people enjoying all the benefits. Yet the rights of the

constituent peoples are realized in cooperation and interdependence, through appropriate participation in institutions of public authority in the decision-making process, while this also constitutionally legitimizes differences within the unified constitutional order, which in a practical sense can represent an obstacle to its stability and integration of the *demos*.

Ulrich Preuss also distinguishes two basic conceptions of nations "while in the French concept the nation is the whole of the *demos*, in the German and Eastern European concepts the nation is a group defined in terms of ethnicity: the nation is the *ethnos*." The constitutional values that make up the constitutional identity, although different from the national identity, notes Rosenfield, still require the latter to create an imaginary community. These two imaginary communities must combine points of convergence and divergence. Moreover, the constitutional identity in question must process and reprocess the material in order to promote a vision that integrates the *ethnos* and *demos* in a constitutionally sustainable way (Rosenfeld 2022, p. 1).

Numerous constitutional law theorists agree that the civic concept of constitutional identity, from the prism of constitutionalism and democracy, is superior to the ethnocentric one (Marinković 2018, p. 99) and is one of the most important determinations of the character of the state of Montenegro. It directs Montenegro not to root its constitutional identity in ethnic origin, but in the *demos* and the normative foundations that Montenegro is "culturally, ethnically, and religiously neutral, i.e., as a state open to all ethnic groups, religions, and cultures, except for those who might violate these basic principles." Such a concept, which is normatively an integral part of the constitutional identity of Montenegro, is contrary to any form of ethnocentrism, it does not oblige anyone to declare their nationality, but implies that nationality is a private human characteristic.

Normative determinations based, in a theoretically superior way, on the civic and multicultural constitutional identity and equality and freedom of all citizens, while preserving their uniqueness and identity, in the current constitutional development of Montenegro represent an ideal type, a desirable society rather than a reality. The dilemmas from the time of the making of the constitution are also present in the post-constitutional period, and today they are modernized in that being which is the Montenegrin identity, through the issue of relations between the state of Montenegro and the Serbian Orthodox Church. Montenegro continues to remain a deeply divided society whose political culture makes it impossible for all citizens to jointly build and jointly accept a civil constitutional identity, because polarization based on national identity suits politicians in order to homogenize their electorate; in this way, the religion, although based on ecumenical principles, and the state, although based on a civic model, are being misused for the purpose of a "power struggle" of political actors that will determine the final result.

The principle of secularity and the values of the civil state are an essential determinant for solving the current crisis arising from the relationship between the state and religious communities. Tolerance and multiculturalism teach us to evaluate people not from the aspect of identity, religious rituals, or ideology, but through their effective activity, which forms the ethical basis of society (Aslan 2019, p. 441). The fact that stratification is one of the main characteristics of identity was also concluded in a study conducted within the framework of the Council of Europe, where the ideal citizen is considered to be one who supports a multilingual and multicultural European community, who understands and promotes the democratic values of solidarity, tolerance, mutual understanding, and respect (Paige 2020, p. 59). Creating one's own identity is possible through understanding the identity of others, i.e., those who are different. Empirical research has shown that in the educational system, teaching about religions is mainly reduced to their similarities and common features, although from a pedagogical point of view it is not acceptable to ignore differences and consider only what is the same or similar in different religions (Kolb 2021, p. 153). There is a very interesting result from research conducted in the Netherlands in 2000, in which the motive for enrolling in religious schools was examined. A large proportion of the respondents, more than 40%, cited the quality of education received, the availability of education, and the reputation of the school as the main reasons for enrolling

(Kienstra et al. 2019, p. 595). Furthermore, the same survey showed a very interesting fact that the largest percentage of parents of students who are enrolled in religious schools declare themselves as non-believers (Kienstra et al. 2019, p. 595). Bearing this in mind and from this distance, we cannot agree with the hypothesis set out in the social sciences of the eighteenth and nineteenth centuries that education has a secularizing effect on the religiosity of students and that believers are generally averse to traditional educational activities (Bertrand 2015, p. 1). Therefore, tolerance and interdependence teach us that we should not treat differences as incompatible opposites (Kolb 2021, p. 153), but rather as quite the contrary.

This leads to the conclusion that it is a completely legitimate goal that all citizens of Montenegro, regardless of nationality, religion, or other individual characteristic, have the right to choose the denominators of their own identification, and that there is also a legitimate right of the Montenegrin Orthodox community to restore the autocephaly of the religious community that recognizes its ethno-national identity, but the address for solving this issue in a secular and civil state is predominantly not the state but rather the church. Therefore, the initiative for a certain degree of autonomy or eventual autocephaly can only come from the religious community and its believers. This is precisely the role of these values in resolving the issue of the relationship between the state and religious communities. If, as pointed out, the constitutional identity of Montenegro rests on the clearly defined principles of a civic and multicultural state and the separation of religious communities from the state, then any arbitrary interference by the state in the internal organization of religious communities, and thus in the organization of the Serbian Orthodox Church, would call into question the secular nature of the state, its healthy sovereignty, and its civic constitutional identity. State authorities have a constitutional duty to ensure religious freedom for all in a spirit of pluralism and mutual tolerance, just as religious communities, although free to carry out their religious affairs and religious ceremonies, have a duty to function within the framework of the national constitutional order of the state in which they perform these religious affairs and ceremonies.

## 5. Conclusions

It is noted that the constitutional and democratic values of Montenegro as a civic, multicultural, and secular state, although they received the legitimacy of a qualified majority of the constitution-making body in the process of adopting the 2007 Constitution, remain unfinished concepts, which are transformed in accordance with historical and social trends. The foundation of constitutional identity embodied in the values of the civic community in the current constitutional and legal development of the state is shaken by the creation of a normative framework and contractual obligations that are not inherent to the secular character of the state.

The legal and political relationship between the state of Montenegro and the Serbian Orthodox Church is not a formally and legally completed process. Numerous disputed issues arising from this relationship should be addressed through an institutional framework, without losing sight of the fact that the main and basic role of all state authorities is to contribute to the development of civic awareness and collective self-awareness that constitutional identities of democratic values remain the only point of unity by which politically mature and civilized communities are known.

The existing relationship indicates that, despite the constitutional organization of Montenegro as a civic and multicultural state, ethnicity and religious affiliation still remain the prevailing political rhetoric and denominator of identity in reality, and the increasing clericalization of society causes the constitutionally determined separation between the state and religious communities to not be always or completely respected.

The whole concept of civic constitutional identity, even in its earliest steps, appears as a complete opposite to the medieval religiously centered culture as a whole. At the core of civic identity is secularization aimed at the prosperity of truth, science, democratic life, and spirit. It is clear that the key constitutional aspiration of the constitution maker to give the

constitutional determination of Montenegrin constitutional identity in the form of a secular and civic state, in order to become a constitutional and political reality, is *de jure* and *de facto* shaken by the increasingly extensive cooperation between the spiritual and the secular. The main seal of constitutional identity is given by the constitutional courts as crystallizers of the state's constitutional values, and the future of the relationship between the state and religious communities will mostly depend on the decisions of the Constitutional Court, and constitutional identity, as a set of values on which the existence and duration of a legally organized community is based, is the supreme vertical that both parties must respect when meeting in the public sphere. Given that the state acts as an equal partner when regulating matters of common interest with religious communities, and not as an executor of the intentions of any religious community, its main task is to protect the constitutional identity of the state, i.e., the basic values and principles of the constitutional order when coming into contact with the applicable Law of Religious Organization, which is limited to its internal space. At the moment of the permissible encounter between the "secular" and the "spiritual" in the public sphere, the spiritual must be held according to the laws of the secular authority.

In practice, the fine line between exceeding the authority of the state and the positive obligations of the state in solving the issue of religious communities is very easy to cross, and the European Court of Human Rights suggests that "state authorities must be vigilant in this extremely delicate area" (Judgment of the European Court of Human Rights, Hasan and Chaush v. Bulgaria, petition number: 30985/96, dated 26 October 2000). It is concluded that in matters of religion one should always consider the broader context, which concerns the legal tradition of each state and the significance that religion has in a particular society, and the specific time and spatial framework in which all these processes take place.

**Author Contributions:** Conceptualization, A.V. and B.D.; methodology, A.V. and B.D.; investigation, A.V. and B.D.; writing—original draft preparation, A.V. and B.D.; writing—review and editing, A.V. and B.D. All authors have read and agreed to the published version of the manuscript.

**Funding:** The APC was funded by Constitutional Court of Montenegro.

**Data Availability Statement:** The data presented in this study are available on request from the corresponding author.

**Conflicts of Interest:** The authors declare no conflict of interest.

## Note

1　　Constitution of Montenegro, *Official Gazette of Montenegro*, 1/2007.

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
