# Peer review of "Uses and Abuses of Religion in the Contemporary Legal Development of Montenegro: Undermining the Principle of Secularity"

_religions, doi:10.3390/rel14040532_

Round 1

Reviewer 1 Report

The thesis for this article is not clear. The assertion that there are influences by the Orthodox Church on the development of the Montenegro government is not strongly argued. There are general discussions with references to other European governments but there are no concrete evidence given of people, events or activities within Montenegro itself. Here, a brief historical excursus may be helpful. Or, what is the attitude of the current president of the Montenegro--Dukanovic? Would this influence how people will react or participate? This can include also a brief history of the religious makeup of the countries involved in the area. It was ruled by the Ottoman Empire while also negotiating the influence of Western Christianity.

The reference to the different Orthodox churches is appropriate but it doesn't clarify which one is directly involved in the government of Montenegro itself nor discuss the wider support from other Orthodox churches. Also, a clarification of the configuration of Eastern Orthodoxy needs to be made. It deserves more than just a brief mention, especially since this is an important part of the argument. 

The abstract does state that the approach of this article will include sociological and axiological methods. Both of these rely on evidence and data to support an argument. There doesn’t seem to be any data about events, people or activities that would persuade a reader of the influence of religion in the government of the state.  There is no way to make a judgment of the misuses of religion in government or vice versa. 

Author Response

We appreciate very much your comments. In regard to your suggestion that extensive English revision should be done, we kindly inform you, that translation from our native language to English was done by the Center for education, translation and publishing. Additionally, after your comments, we have also done the proofreading by a lector-native speaker.

After your comments regarding the attitude of the current President of Montenegro, Mr. Djukanovic we have extended our manuscript with brief overview of the events in last two and the half years that you can find in our final version of the text in track changes.

Furthermore, we have included case law on the principle of secularity of the surrounding states which share similar legal heritage as Montenegro.

We have also included more references in our bibliography.

Reviewer 2 Report

This article provides an insightful analysis of the relationship between the state of Montenegro and the Orthodox Church in the context of constitutional legal development, multiculturalism, and tolerance. The author effectively highlights the significance of religion in contemporary society and acknowledges the complexities associated with establishing boundaries between the state and religious communities.

The author presents a well-structured argument, using examples from post-socialist regimes and the United States, to demonstrate the immanent interdependence of religion and politics in transitional and democratic societies. The author's hypothesis regarding the shaken civil and secular constitutional identity in Montenegro is well-supported with evidence and is relevant to the current political climate.

The article is well-written and demonstrates a good command of language. The author effectively uses relevant terminology and concepts in the field of law, religion, and politics to support their argument. The article could be improved by providing more specific examples of how the Orthodox Church directly influenced the outcome of parliamentary elections in Montenegro, as well as discussing the potential implications of this influence on the legal relationship between the state and religious communities.

Overall, the article is a valuable contribution to the field of law, religion, and politics, and is suitable for publication in an academic journal. The author's analysis and insights are well-reasoned, and the article is well-written and well-structured.

Author Response

Thank you very much on your meaningful comments.

In regard to your suggestions we extended our manuscript and provided the information on how the Serbian Orthodox Church directly influenced the outcome of parliamentary elections in Montenegro, together with case-law of the Constitutional Court of Montenegro and surrounding states.

We have also included new references in our bibliography.  

Round 2

Reviewer 1 Report

The addition of the explanation of the constitution of the country provides good reason for the thesis posed in this article. The elaboration of the Orthodox Church's influence in the country is also helpful. These additions have made the article more substantial and easier to understand. It gives the reader a reason to be interested in the article.